# Improved Thermoelectric Performance of Sb_2_Te_3_ Nanosheets by Coating Pt Particles in Wide Medium-Temperature Zone

**DOI:** 10.3390/ma16216961

**Published:** 2023-10-30

**Authors:** Qing-Ling Guan, Li-Quan Dong, Qun Hao

**Affiliations:** 1Beijing Institute of Technology, School of Optics & Photonics, Beijing Key Laboratory Precise Optoelectronics Measurement Institute, Beijing 100081, China; guanql@bit.edu.cn (Q.-L.G.); qhaoprof@126.com (Q.H.); 2Beijing Institute of Technology, Yangtze Delta Region Academy, Jiaxing 314019, China

**Keywords:** thermoelectric materials, Sb_2_Te_3_ nanosheets, Pt nanoparticles, solvothermal method

## Abstract

The p-type Sb_2_Te_3_ alloy, a binary compound belonging to the V_2_VI_3_-based materials, has been widely used as a commercial material in the room-temperature zone. However, its low thermoelectric performance hinders its application in the low-medium temperature range. In this study, we prepared Sb_2_Te_3_ nanosheets coated with nanometer-sized Pt particles using a combination of solvothermal and photo-reduction methods. Our findings demonstrate that despite the adverse effects on certain properties, the addition of Pt particles to Sb_2_Te_3_ significantly improves the thermoelectric properties, primarily due to the enhanced electronic conductivity. The optimal *ZT* value reached 1.67 at 573 K for Sb_2_Te_3_ coated with 0.2 wt% Pt particles, and it remained above 1.0 within the temperature range of 333–573 K. These values represent a 47% and 49% increase, respectively, compared to the pure Sb_2_Te_3_ matrix. This enhancement in thermoelectric performance can be attributed to the presence of Pt metal particles, which effectively enhance carrier and phonon transport properties. Additionally, we conducted a Density Functional Theory (DFT) study to gain further insights into the underlying mechanisms. The results revealed that Sb_2_Te_3_ doped with Pt exhibited a doping level in the band structure, and a sharp rise in the Density of States (DOS) was observed. This sharp rise can be attributed to the presence of Pt atoms, which lead to enhanced electronic conductivity. In conclusion, our findings demonstrate that the incorporation of nanometer-sized Pt particles effectively improves the carrier and phonon transport properties of the Sb_2_Te_3_ alloy. This makes it a promising candidate for medium-temperature thermoelectric applications, as evidenced by the significant enhancement in thermoelectric performance achieved in this study.

## 1. Introduction

Thermoelectric materials have garnered significant attention as a promising and sustainable source of new energy due to their ability to directly convert heat into electricity through the Seebeck effect [1,2,3,4,5,6]. The performance of thermoelectric devices is commonly evaluated using the dimensionless figure of merit (*ZT*, *ZT* = *S*^2^*σT*/*κ*) where *S*, *σ*, *T*, and *κ* represent the Seebeck coefficient, electrical conductivity, absolute temperature in Kelvin, and total thermal conductivity, respectively [7,8]. The thermoelectric efficiency is directly proportional to the *ZT* value. A higher *ZT* value indicates a more efficient conversion of heat into electrical energy, leading to a higher thermoelectric efficiency [9]. The *ZT* value is a figure of merit that quantifies the thermoelectric efficiency of a material. It is an important parameter in assessing the performance of thermoelectric materials. The *ZT* value allows researchers and engineers to compare different materials and identify those with the highest thermoelectric efficiency. It serves as a crucial parameter in the development and optimization of thermoelectric materials for applications such as waste heat recovery, power generation, and solid-state cooling [10]. To achieve excellent thermoelectric properties and high efficiency, it is crucial to minimize thermal conductivity (*κ*) while maximizing the power factor (*PF*, *PF* = *S*^2^*σ*) in order to obtain a higher *ZT* value [11,12]. However, achieving substantial *ZT* values remains an immense challenge due to the intricate interplay between the Seebeck coefficient, electrical conductivity, carrier concentration, and electronic structure [13,14]. Total thermal conductivity is a key parameter in thermoelectric materials, comprising electronic thermal conductivity (*κ_e_*) and lattice thermal conductivity (*κ_l_*). *κ_e_* relates to heat conduction by charge carriers, while *κ_l_* pertains to heat conduction through lattice vibrations [15,16]. In thermoelectric materials, the goal is to maximize electrical conductivity and minimize electronic thermal conductivity to retain heat. Additionally, reducing lattice thermal conductivity through structural modifications inhibits heat transfer, maintaining a significant temperature difference. Optimizing total thermal conductivity enhances the *ZT* and improves energy conversion efficiency for practical applications like waste heat recovery and energy-efficient cooling.

Thermoelectric devices are commonly designed and optimized for operation within the following temperature ranges: low-temperature zone (<200 °C); medium-temperature zone (200–600 °C); high-temperature zone (>600 °C) [4]. In this paper, we demonstrated that Sb2Te3 coated with 0.2 wt% Pt particles can maintain *ZT* values above 1.0 within the temperature range of 333–573 K (60–300 °C). Some examples of room temperature thermoelectric applications include: energy harvesting (low-power electronic devices, sensors, or wireless communication modules, eliminating the need for batteries or extending their lifetime); climate control (such as in portable refrigeration, cooling seats in vehicles, or small-scale air conditioning for electronic enclosures); thermal management (manage heat dissipation and maintain stable operating temperatures, improving the reliability and performance of electronic components) [17]. Some examples of medium-temperature thermoelectric applications include: waste heat recovery (recover waste heat from high-temperature industrial processes, such as exhaust gases from power plants, furnaces, or industrial manufacturing plants); geothermal energy (use Earth’s heat to generate electricity); solar power generation (convert solar heat into electricity) [18].

V_2_VI_3_-type (V = Bi, Sb; VI = Te, Se) thermoelectric materials have gained significant attention due to their highly desirable thermoelectric properties. As a benchmark, both p-type and n-type Bi_2_Te_3_-based alloys demonstrate a *ZT* value of approximately one at room temperature, highlighting their excellent performance in converting heat differentials into electrical energy [19,20,21,22,23,24,25]. However, their application in the medium-temperature zone is restricted by the inherent characteristics of narrow band gap compounds. Various processing techniques have been employed to improve the power factor or reduce thermal conductivity, including nanotechnology (energy barrier filtering [26], minority carrier blocking [27], liquid-phase hot deformation [28], textured engineering [29]) and band engineering (resonant level [30], band convergence [31], electronic density of states distortion [32]).

Zheng et al. [33] demonstrated that by controlling the geometric sizes and introducing nano-sized Au particles, the relative contributions of electrons can be adjusted, leading to enhanced phonon scattering. This approach has been applied to improve the thermoelectric properties of Sb_2_Te_3_ nanosheets, resulting in a *ZT* value of 0.8 at 523 K. Unlike solvothermal methods that form larger metal particles, which greatly influence thermal conductivity, the synthesis of small-sized particles is necessary to primarily modify the electrical transport properties. Furthermore, DFT studies have proven to be effective in analyzing the thermoelectric properties of materials [34,35].

In this study, we aimed to enhance the thermoelectric performance of Sb_2_Te_3_ by developing a composite material consisting of Sb_2_Te_3_ hexagonal plates coated with Pt nanoparticles. The composite was synthesized using a solvothermal method and a photo-reduction method. By incorporating Pt nanoparticles, we successfully constructed a carrier channel within the material, resulting in a significant improvement in its electrical conductivity. Remarkably, this enhancement in electrical conductivity was achieved while maintaining the thermal conductivity of the material at nearly unchanged levels. As a result, the overall thermoelectric performance of the Sb_2_Te_3_-based composite was greatly improved. Specifically, the Sb_2_Te_3_ composite, incorporating 0.2 wt% Pt particles, exhibited a maximum electrical conductivity (*σ_max_*) of 1.38 × 10^5^ S m^−1^ at 300 K. Furthermore, the composite demonstrated an impressive maximum figure of merit (*ZT_max_*) value of 1.67 at 573 K. Sb_2_Te_3_ coated with 0.2 wt% Pt particles can maintain *ZT* values above 1.0 within the temperature range of 333–573 K (60–300 °C). This temperature zone covers a 200 K range (low-temperature zone and medium-temperature zone), making it suitable for a broader range of thermoelectric applications, including waste heat recovery and automotive exhaust systems. By operating within this wider temperature zone, thermoelectric devices can effectively harness waste heat and improve energy efficiency in various industrial and automotive settings.

## 2. Experiment

### 2.1. Synthesis of Sb_2_Te_3_ NCs

Potassium Tellurite (K_2_TeO_3_, 99.5%), antimony trichloride (SbCl_3_, 99.9%), and chloroplatinic acid (H_2_PtCl_6_) were purchased from Aladdin Company in Shanghai, China. Sodium hydroxide (NaOH, ≥98%), isopropanol ((CH_3_)_2_CHOH, ≥98%), absolute ethanol (CH_3_CH_2_OH, ≥98%), diethylene glycol (DEG, HOCH_2_CH_2_OCH_2_CH_2_OH, 99%), and polyvinylpyrrolidone (PVP, (C_6_H_9_NO)_n_, average mol wt ∼55,000) were purchased from Sinopharm Group. Acetone (CH_3_COCH_3_, ≥98%) was obtained from various sources. All chemicals were used without further purification. Firstly, 4 mmol SbCl_3_, 6 mmol K_2_TeO_3_, 10 mmol KOH, and 0.15 g PVP were dissolved at room temperature in 80 mL DEG in a 250 mL three-necked flask for 10 min. Then, under the action of magnetic stirring, slowly heated to 250 degrees Celsius for 6 h, during which DEG was added as a high boiling solvent and reducing agent [36], and PVP was added as a surface modifier to the solvent [37]. The final product Sb_2_Te_3_ hexagonal nanosheets were obtained from black solution by high-speed centrifugation. Subsequently, chloroplatinic acid was dissolved in deionized water to form a uniform solution with a mass solubility of 20 mg/mL. Next, the synthesized Sb_2_Te_3_ hexagonal tablets were dissolved in deionized water in a 100 mL beaker and then uniformly dispersed using ultrasound. Finally, a uniform solution containing Pt^4+^ ions was added in proportion to the aforementioned beaker. Among them, Pt was generated by photoreduction method. Under traditional Xe light irradiation, platinum chlorate was reduced to generate Pt, and the specific reaction was as follows:(1)$$H_{2}{\mathrm{PtCl}}_{6}\to 2H^{+}+{\mathrm{PtCl}}_{6}^{2-}$$
(2)$${\mathrm{PtCl}}_{6}^{2-}+4e^{-}\to \mathrm{Pt}+6{\mathrm{Cl}}^{-}$$

### 2.2. Synthesis of Sb_2_Te_3_ NCs Coated Pt Nanoparticles

Chloroplatinic acid was dissolved in deionized water to form a uniform solution with a mass solubility of 20 mg/mL. The synthesized Sb_2_Te_3_ hexagonal sheets were dissolved into 100 mL beaker with deionized water, and then were dispersed uniformly by ultrasound. The dissolution and subsequent dispersion of the sheets in our synthesis process serve two purposes. Firstly, it allows the platinum ions to adhere more effectively to the surface of Sb_2_Te_3_, ensuring good interaction between the two materials. This improved adhesion helps to enhance the overall performance of the composite material. Secondly, the dissolution and dispersion process contribute to achieving a more uniform distribution of platinum nanoparticles on the surface of Sb_2_Te_3_. This uniformity is important for obtaining consistent and reliable properties throughout the material. By dissolving the sheets and dispersing them, we create conditions that facilitate the homogeneous distribution of platinum nanoparticles, leading to improved conductivity and other desired properties. A homogeneous solution containing Pt^4+^ ions was added to the above beaker in proportion. Pt nanoparticles were reduced and were attached to the surface of Sb_2_Te_3_ hexagonal sheet through traditional Xe light-reduction method under full spectrum conditions simultaneously. Then, the Sb_2_Te_3_-coated Pt nanoparticles were washed and collected by high-speed centrifugation. The final product was dried at 60 degrees for 6 h. These fine powders were consolidated to ensure high density under argon atmosphere by spark plasma sintering (SPS) under a pressure of 50 MPa at 523 K for 10 min. The surface carbon paper of sintered sample was removed through sandpaper of different meshes. Subsequently, the sintered bulk was cut into different sizes to meet different measurements.

### 2.3. Characterization and Thermoelectric Measurements

The phase composition of all nanocomposite specimens were investigated by powder X-ray diffraction analysis (XRD; SmartLab, Rigaku Corporation, Tokyo, Japan, with Cu Kα radiation, *λ* = 1.5406 Å). The scanning range of 2θ was 10~90 degrees, the scan speed was 6 degrees per minute, and the scan increment was 0.02 degrees. The lattice parameters of all samples were calculated by the Rietveld refinement method. The microstructural properties and morphology were further investigated by field emission scanning electron microscopy (FE-SEM) equipped with the X-ray energy dispersive spectroscopy (EDS) and microstructures were fully characterized by high resolution transmission electron microscopy (HRTEM) (JEOL-F200 instrument, with an acceleration voltage of 200 kV). Electrical conductivity (σ) and Seebeck coefficient (*S*) were attained simultaneously on a commercial system (LSR-3, Linseis, Bavaria, Germany). The thermal diffusivity coefficient (*D*) were measured for the disk sample on a laser flash system (Netzsch LFA 457, Netzsch, Selb, Germany). The heat capacity (*C_p_*) was assumed to be the Dulong–Petit limit and to be temperature independent. The sample density (*ρ*) was determined by the Archimedes method, and then the total thermal conductivity was calculated using the formula of κ = DρC_p_. Hall coefficient (*R_H_*) was determined under a magnetic field of 6000 G by Van der Pauw method and the carrier concentration (n_H_ = 1/(eR_H_)) of specimens was calculated. Then, the Hall carrier mobility (*μ_H_*) was calculated by σ = eμ_H_n_H_. The accuracies in measurements of electrical conductivity and Seebeck coefficient are ±2% and ±6%, respectively. The accuracies in measurements of thermal diffusivity coefficient and sample density are ±5% and ±3%, respectively. Finally, the uncertainty of *ZT* is around ±10%.

### 2.4. DFT Calculations

The first-principles density functional theory calculations are implemented in the Quantum ESPRESSO (QE) code [38,39] by using projected augmented wave (PAW) [40] and Perdew–Burke–Ernzerhof (PBE)-type generalized gradient approximation (GGA) [41]. The corresponding pseudopotential files are taken from the SSSP PBEsol Precision v1.3.0 [42]. The G-centered Monkhorst–Pack k-point mesh [43] is considered as 12 × 12 × 1. The spin–orbit coupling (SOC) has been taken into consideration. To obtain stable structures, the energy and force convergence thresholds were set to 10^−6^ Ry and 10^−2^ Ry. The electronic transport properties are calculated by the BoltzTraP code [44] based on Boltzmann transport theory. The views of structure and charge density are illustrated by VESTA software (3.5.7) [45].

## 3. Result and Discussion

Figure 1a,b displays the X-ray diffraction (XRD) patterns of Sb_2_Te_3_ and composite specimens containing Sb_2_Te_3_ coated with various weight percentages (x wt%) of Pt particles (x = 0, 0.2, 0.4, and 0.6). The main XRD peaks of all specimens correspond well with the standard PDF cards of Sb_2_Te_3_ (PDF#71-0393). Notably, a weak peak at the angle of 27°, which corresponds to Te (PDF#86-2269), is conspicuously absent in the X-ray diffraction patterns of all the specimens. This absence indicates that the excess Te, resulting from the deliquescence of antimony trichloride, appears as a second phase, as illustrated in Figure 1a,b. The deliquescence of antimony trichloride resulted in the formation of excessive Te as a second phase. This had an impact on the characterization analysis. Additionally, the uniformity of the Pt coating needs improvement, which may be related to the uniform dispersion of Sb_2_Te_3_ in the solution during the photoreduction process.

The XRD patterns of Sb_2_Te_3_ coated with Pt demonstrate no additional second phase, indicating that the presence of Pt nanoparticles does not alter the crystal structure of Sb_2_Te_3_. To quantitatively analyze the actual Te content, Rietveld refinements for the pure specimen were performed using the Generalized Structural Analysis System (GSAS-II), as depicted in Figure 1c. Concurrently, the lattice parameters a and c were determined for all samples through Rietveld refinements, as shown in Figure 1d. Among all of the specimens, the lattice parameter a of Sb_2_Te_3_ coated with 0.2 wt% Pt is higher, while the lattice parameter c is lower. It is important to note that the presence of Te precipitation in the second sample with relatively different Te content may introduce some errors, but it does not affect the overall experimental results.

The XRD analysis confirms that the composite specimens maintain the crystal structure of Sb_2_Te_3_, with the excess Te appearing as a second phase due to the deliquescence of antimony trichloride. The presence of Pt nanoparticles does not introduce any additional second phase. The quantitative analysis of Te content and lattice parameters further supports these findings, with the Sb_2_Te_3_ coated with 0.2 wt% Pt exhibiting distinctive values for lattice parameters a and c compared to the other specimens.

To gain a comprehensive understanding of the enhanced thermoelectric (TE) properties, a detailed microstructural analysis was conducted using Field Emission Scanning Electron Microscopy (FE-SEM) and High-Resolution Transmission Electron Microscopy (HRTEM) on Sb_2_Te_3_ coated with Pt nanoparticles.

FE-SEM observations revealed the morphology of pure Sb_2_Te_3_, as shown in Figure 2a, exhibiting a regular hexagonal sheet structure. The SEM micrographs further revealed that the Sb_2_Te_3_ nanosheets had dimensions of approximately 8 μm with a thickness of about 500 nm, as depicted in Figure 2d. In order to assess the uniformity, element mapping of Sb and Te was performed using Energy-Dispersive X-ray Spectroscopy (EDS), as shown in Figure 2b,c, confirming the good homogeneity of the material.

Figure 2e displays the morphology of Sb_2_Te_3_ coated with 0.2 wt% Pt nanoparticles. The surface of the hexagonal pieces exhibited a transition from a smooth state to a rough state, indicative of the presence of Pt nanoparticles. To obtain detailed information regarding the specific composition of the surface particles, HRTEM analysis was conducted at an acceleration voltage of 200 kV. Under low magnification conditions, the hexagonal sheet structure of the Sb_2_Te_3_ nanomaterials remained intact, as shown in Figure 3b. Element mapping of Sb and Te, depicted in Figure 3c,d, respectively, highlighted the homogeneity of the nanosheets. Moreover, element mapping of Pt within the same region confirmed the uniform coating of Pt particles on the surface of the hexagonal sheets, as illustrated in Figure 3e and Table 1.

The microstructural analysis conducted using FE-SEM and HRTEM provided valuable insights into the morphology and composition of Sb_2_Te_3_ coated with Pt nanoparticles. The observations confirmed the regular hexagonal sheet structure of Sb_2_Te_3_ and demonstrated the uniformity of the material. The presence of rough surfaces and the uniform distribution of Pt nanoparticles were observed, further supporting the enhanced TE properties observed in the composite material. SEM analysis confirmed successful synthesis of Sb_2_Te_3_ coated with 0.2 wt% Pt nanoparticles, uniformly adhered to the surface. The presence of Pt atoms enhanced electronic conductivity, promoting charge carrier transport. TEM observations confirmed a hexagonal sheet-like structure and material uniformity. Rough surfaces and a uniform distribution of Pt nanoparticles were observed. Pt presence significantly increased carrier concentration, enhancing conductivity.

The presence of ${Sb}_{Te}^{\prime }$ antisite defects in Sb_2_Te_3_ nanosheets, synthesized using the solvothermal method, contributes to the generation of holes according to Equation (3). This leads to the material exhibiting p-type conductivity [46].
(3)$${\mathrm{Sb}}_{2}{\mathrm{Te}}_{3}\to {\mathrm{Sb}}_{\mathrm{Te}}^{\prime }{+2\mathrm{Te}}_{\mathrm{Te}}^{\times }{+\mathrm{Sb}}_{\mathrm{Sb}}^{\times }{+\mathrm{Te}}_{\left(g\right)}+h^{⦁}$$

In Figure 4a, the relationship between the Seebeck coefficient and temperature for Sb_2_Te_3_ coated with Pt is presented. The measured Seebeck coefficient (*S*) of all Sb_2_Te_3_-coated Pt nanoparticle pellets demonstrates a consistent p-type behavior across the entire temperature range. This indicates that the addition of Pt particles does not alter the conductivity type of the base material. The values of the Seebeck coefficient increase with temperature until they reach a maximum value at a specific temperature (*T* ≈ 523 K). Subsequently, the Seebeck coefficient decreases with further increases in temperature for Sb_2_Te_3_ coated with 0 wt%, 0.4 wt%, and 0.6 wt% Pt within the measured temperature range. However, for Sb_2_Te_3_ coated with 0.2 wt% Pt, the Seebeck coefficient (*S*) exhibits a monotonic increase with temperature. The incorporation of Pt particles in Sb_2_Te_3_ does not change its p-type conductivity. The Seebeck coefficient shows temperature-dependent behavior, with different patterns observed for different Pt concentrations. The monotonic increase in the Seebeck coefficient with temperatures for Sb_2_Te_3_ coated with 0.2 wt% Pt suggests a unique thermoelectric response in that specific composition.

Based on the Mott formula [47,48,49,50] for a degenerated semiconductor, the Seebeck coefficient could be written as follows:(4)$$S=\frac{\pi^{2}k_{b}^{2}T}{3e}{\left[\frac{\partial \ln(\sigma (E))}{\partial E}\right]}_{E=E_{F}}=\frac{\pi^{2}k_{b}^{2}T}{3e}{\left[\frac{1}{e}\frac{\partial p(E)}{\partial E}+\frac{1}{\mu }\frac{\partial \mu (E)}{\partial E}\right]}_{E=E_{F}}$$

In relaxation time approximation, the relation between carrier mobility, relaxation time, and energy could be written as $\mu (E)=\frac{e\tau (E)}{m^{*}}$ and $\tau (E)=\tau_{0}E^{\lambda -1/2}$($\tau_{0}$ is a constant independent of parameter (*λ*)), then Mott Equation (4) could be expressed as
(5)$$S=\frac{\pi^{2}k_{b}^{2}T}{3e}{\left[\frac{1}{e}\frac{\partial n(E)}{\partial E}+\frac{\lambda -1/2}{E}\right]}_{E=E_{F}}$$

Under conditions where acoustic phonon scattering is the major scattering mechanism, the single parabolic (SPB) model [51,52] is used to study the variation in the effective mass and the relation is written as:(6)$$S=\frac{8\pi^{2}k_{b}^{2}}{3eh^{2}}m^{*}T{\left(\frac{\pi }{3n}\right)}^{2/3}$$
where *S* is Seebeck coefficient, *k_b_* is Boltzmann’s constant, *e* is the electron charge, *h* is the Planck constant, *m** is the effective mass, *T* is the temperature in Kelvin, and *n* is carrier concentration.

In order to investigate the variations in electrical transport properties and effective mass, measurements of carrier concentration, carrier mobility, and effective mass were conducted using the Van der Pauw method [53]. The results are summarized in Table 2. All samples exhibit p-type electrical transport behavior, consistent with the observed Seebeck coefficient. The addition of Pt particles has a notable effect on the carrier mobility (*μ*) at 300 K. The carrier mobility values decrease from 4.63 × 10^2^ cm^2^/Vs, 3.63 × 10^2^ cm^2^/Vs, and 2.79 × 10^2^ cm^2^/Vs to 2.5 × 10^2^ cm^2^/Vs with the incorporation of Pt particles. The decrease in carrier mobility can be attributed to the presence of Pt particles, which act as a second phase and impede carrier transfer.

It is worth noting that the carrier concentration of the composite samples initially increases from 0 to 0.2 wt% Pt, reaching a maximum value, and then decreases with a further increase in Pt particle content. The carrier concentrations of all the samples coated with Pt particles are significantly higher than those of the pure samples. The addition of a small amount of Pt particles effectively reduces the potential barrier between the nano-hexagonal sheets, facilitating carrier transfer. Low-energy carriers can pass through the interfacial potential barrier without annihilation, leading to an increase in carrier concentration.

The effective mass (*m**) of Sb_2_Te_3_ coated with x wt% Pt particles at 300 K was calculated using the measured Seebeck coefficient (*S*) and carrier concentration (*n*) based on Equation (6). These values can vary due to the electronic band structure, impurities, and dopants. The presence of Pt atoms can influence effective mass by altering the band structure, carrier concentration, and scattering mechanisms. Dopants interact with electronic orbitals, causing modifications in effective mass values. In undoped materials, effective mass is determined by the intrinsic band structure, but dopants introduce changes. The effective mass values for Sb_2_Te_3_ nanosheets coated with x wt% Pt particles are 0.2646 m_e_, 0.4691 m_e_, 0.4162 m_e_, and 0.2875 m_e_, respectively. Notably, Sb_2_Te_3_ coated with 0.2 wt% Pt particles exhibits the highest effective mass compared to the other treated samples and the pure sample. This observation is consistent with the increase in electrical conductivity. The measurements of carrier concentration, carrier mobility, and effective mass reveal the impact of Pt particles on the electrical transport properties of Sb_2_Te_3_. The addition of Pt particles decreases carrier mobility but increases carrier concentration. The effective mass of the composite samples varies with the Pt particle content, with Sb_2_Te_3_ coated with 0.2 wt% Pt particles exhibiting the highest effective mass, aligning with the enhanced electrical conductivity observed in the material.

Figure 4b illustrates the variation in electrical conductivity with temperature for all the measured composite specimens. It is observed that the electrical conductivity decreases monotonically with increasing temperature throughout the measured temperature range, indicating that the specimens behave as degenerate semiconductors. However, the introduction of Pt nanoparticles leads to an increase in electrical conductivity. At 300 K, the electrical conductivity increases from 6.46 × 10^4^ S/m to 1.38 × 10^5^ S/m due to the coating of Pt nanoparticles. This increase can be attributed to the variation in carrier concentration. The Pt particles occupy interstitial sites within the nanosheets, acting as conductive channels and enhancing the carrier concentration and electrical conductivity.

Figure 4c presents the thermal conductivity measurements for all Sb_2_Te_3_ samples coated with different concentrations of Pt particles. The excellent thermal transport properties of the composites mainly stem from their textured structure, as depicted in Figure 2f. In comparison to the sample with x = 0 (approximately 0.5346 W/mK at 300 K), the bulk composites containing Pt particles exhibit a slight increase in thermal conductivity. Notably, the sample with x = 0.2 demonstrates the highest *κ*, ranging from 0.56 W/mK to 0.66 W/mK across the entire temperature range of measurement.

Figure 4b demonstrates the degenerate semiconductor behavior of the composite specimens, with electrical conductivity decreasing with increasing temperature. The coating of Pt nanoparticles contributes to an increase in electrical conductivity by enhancing carrier concentration. Figure 4c illustrates the thermal conductivity measurements, with the textured structure of the composites playing a significant role in their excellent thermal transport properties. The addition of Pt particles leads to a slight increase in thermal conductivity, with the sample coated with 0.2 wt% Pt exhibiting the highest thermal conductivity in the measured temperature range. Figure 4d displays the dimensionless figure of merit of the samples at different coated concentrations (x) as a function of temperature. The *ZT* value is a key parameter that indicates the effectiveness of a material for thermoelectric applications, considering the synergetic effects of its electrical and thermal properties. In this study, the *ZT* value exhibits significant improvement.

Remarkably, the *ZT* value remains above 1.0 within the temperature range of 333 K to 573 K. Specifically, for Sb_2_Te_3_ coated with 0.2 wt% Pt particles, the *ZT* value reaches 1.67 at 573 K, which is 47% higher than the *ZT* value of the pure Sb_2_Te_3_ matrix. This enhancement in *ZT* is a result of the combined improvements in electrical and thermal properties achieved by incorporating Pt nanoparticles. Furthermore, the average *ZT* value (*ZT_ave_*) reaches 1.32 within the measured temperature range of 300 K to 573 K. This notable *ZT_ave_* value positions the material as a promising candidate for medium-temperature TE applications. Figure 4d demonstrates the substantial enhancement in *ZT* values achieved by incorporating Pt nanoparticles. The *ZT* value remains above 1.0 over a wide temperature range, with the sample coated with 0.2 wt% Pt particles exhibiting the highest *ZT* value of 1.67 at 573 K. The average *ZT* value of 1.32 further highlights the material’s potential for medium-temperature TE applications.

A small amount of Pt doping has entered the matrix, which prompted us to conduct DFT calculations. Here, we focus on discussing the role of a small amount of Pt doping in the Sb_2_Te_3_ structure, rather than a Pt coating. The study involved constructing Sb_2_Te_3_ samples using a conventional cell with a 2 × 2 × 1 supercell, with the Pt atom positioned in the middle of the interlayer. The structures are visually depicted in Figure 5a. Following structural relaxation, the lattice constant for the Sb_2_Te_3_ conventional cell was determined to be 4.337 Å, while for the Sb_2_Te_3_ 2 × 2 × 1 supercell with Pt, it was found to be 8.685 Å. These lattice constants closely align with experimental results, lending credibility to the study.

Band structure and corresponding Density of States (DOS) analyses were performed on these structures, as depicted in Figure 5b–e. The examination revealed that the Sb_2_Te_3_ conventional cell is a direct semiconductor with a bandgap of 0.15 eV. Both the conduction band minimum and valence band maximum are located at the gamma point. Upon Pt doping, a doping level is observed, with the valence band shifting towards the Fermi energy, indicating p-type doping. Conversely, the conduction band remains largely unaffected. In the DOS plots, a notable sharp increase is observed at around −0.25 eV in Figure 5e. The DOS refers to the number of available energy states per unit volume at a given energy level within the material. When there is a sharp rise in the DOS, it means that there is a higher density of energy states available for electrons to occupy. This can lead to a higher population of charge carriers (holes) in the material, effectively increasing the charge carrier concentration. Overall, the DOS profiles are enhanced due to the presence of Pt, contributing to the improved thermoelectric properties that were observed.

We conducted a theoretical analysis of the electronic transport properties, specifically the Seebeck coefficient and electronic conductivity, as depicted in Figure 6a,b. The carrier concentration for pristine Sb_2_Te_3_ was determined to be 8 × 10^18^ cm^−3^, while for Sb_2_Te_3_ with Pt doping, it increased to 5 × 10^19^ cm^−3^. The electronic relaxation time was set at 10^−13^ s. Notably, the introduction of Pt doping induces the emergence of additional electronic bands, resulting in a substantial improvement in in-plane electronic conductivity while causing a decrease in out-plane electronic conductivity. Regarding the Seebeck coefficient, the values in the in-plane and out-plane directions exhibit close proximity. However, with the addition of Pt, we observed a decrease in the Seebeck coefficient. However, the growth rate of electronic conductivity surpasses that of the Seebeck coefficient, resulting in an overall improvement in the thermoelectric properties. The temperature dependence of these electronic transport properties aligns with experimental observations, reflecting the semiconductor nature of the material. Furthermore, Figure 6c presents a 2D side view of the charge density. It demonstrates that the charge density is localized around the Pt atoms [54]. Moreover, the planar average potential shown in Figure 6d yields similar values throughout, except for regions around the Pt atom. This localization of charge density and the distinct potential values near the Pt atom can be attributed to the enhanced thermoelectric properties observed in the system.

## 4. Conclusions

In summary, a novel approach combining the solvothermal method with the photo-reduction method was employed to prepare p-type Sb_2_Te_3_ nanosheets incorporated with nanometer-sized Pt particles. Then, their thermoelectric transport properties were fully investigated. Those results indicate that nanometer-sized Pt particles can boost electrical conductivity remarkably and slightly improve thermal conductivity. As a result, a Sb_2_Te_3_-based composite system incorporated with 0.2 wt% Pt particles contributes a large *σ_max_* = 1.38 × 10^5^ S m^−1^ at 300 K and *σ_ave_* = 8.7 × 10^4^ S m^−1^. Furthermore, density functional theory (DFT) calculations supported the experimental results by demonstrating that Pt doping induces a sharp increase in the density of states (DOS), with charge density localized at the Pt atoms. This theoretical insight provides a deeper understanding of the underlying mechanisms behind the enhanced electrical conductivity observed in the composite system. Finally, a large *ZT_max_* = 1.67 (at 573 K) and *ZT_ave_* = 1.32 (in the measured temperature range of 300~573 K) are reached. The incorporation of nanometer-sized Pt particles into p-type Sb_2_Te_3_ nanosheets resulted in a substantial enhancement of electrical conductivity and a minor improvement in thermal conductivity. This, in turn, led to the development of a composite material with high electrical conductivity and superior thermoelectric performance. The combination of experimental results and theoretical insights underscores the efficacy of metal particle optimization in achieving enhanced electron and phonon transport properties, ultimately improving the thermoelectric performance of the material.

## Figures and Tables

**Figure 1 materials-16-06961-f001:**
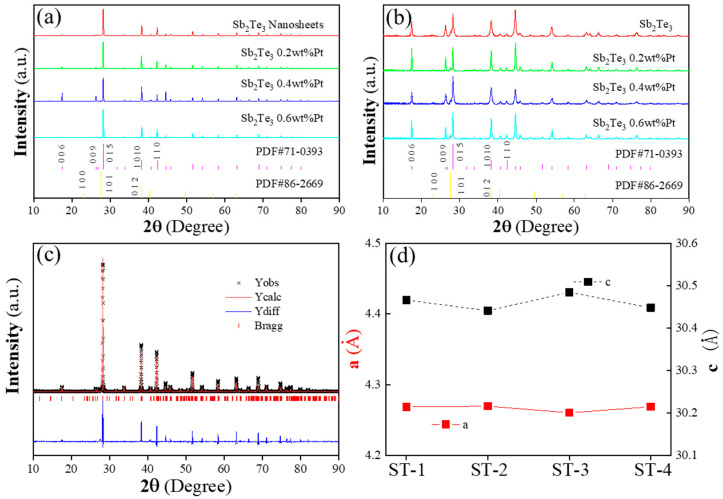
(**a**) XRD patterns of Sb_2_Te_3_ powders. (**b**) XRD patterns of Sb_2_Te_3_ bulks after SPS. (**c**) Refined patterns of Sb_2_Te_3_ powders (**d**) Lattice parameter a and c.

**Figure 2 materials-16-06961-f002:**
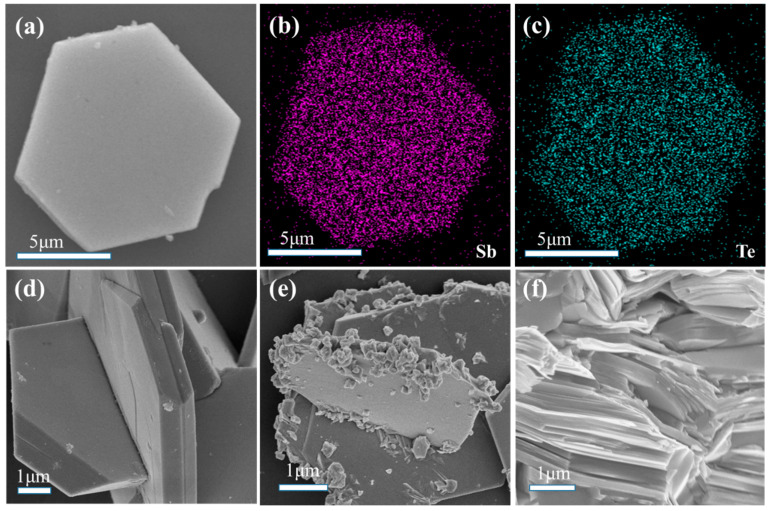
(**a**) FESEM micrograph of pure Sb_2_Te_3._ (**b**,**c**) EDS mappings. (**d**) Thickness of Sb_2_Te_3_ nanosheets. (**e**) Sb_2_Te_3_-coated 0.2 wt% Pt nanoparticles. (**f**) Fragment of Sb_2_Te_3_ bulk.

**Figure 3 materials-16-06961-f003:**
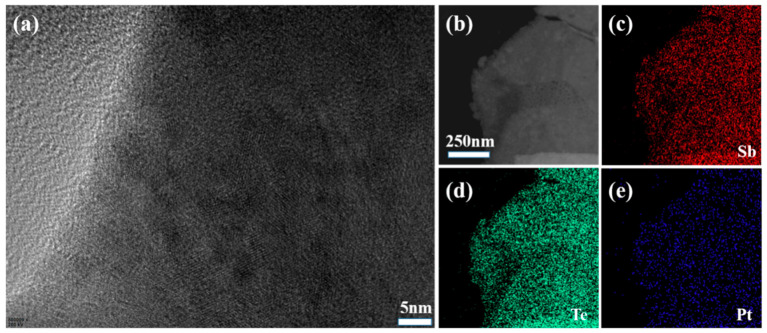
(**a**) High-magnification TEM image of Sb_2_Te_3_ coated Pt nanoparticles. (**b**) Low magnification TEM image of Sb_2_Te_3_ coated Pt nanoparticles. (**c**–**e**) Corresponding EDs elemental mapping images of Sb, Te, Pt.

**Figure 4 materials-16-06961-f004:**
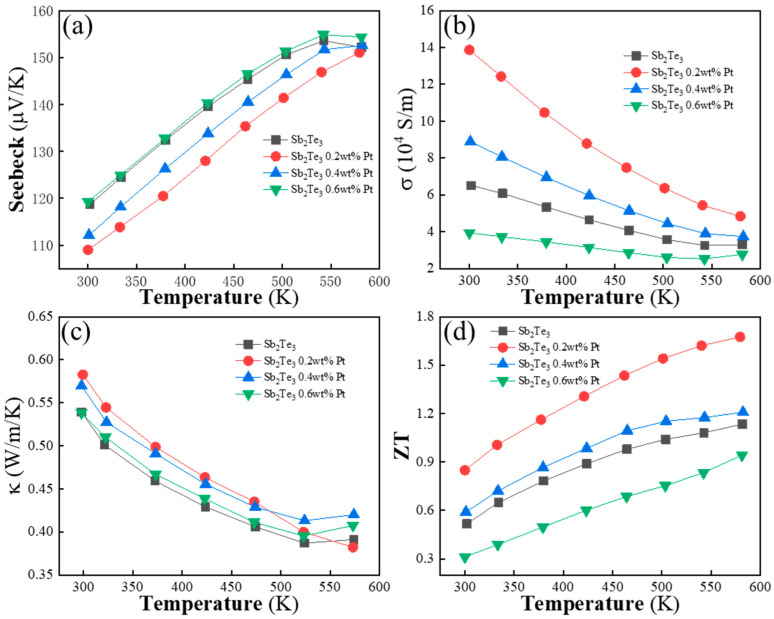
(**a**) Seebeck coefficient vs. Temperature. (**b**) Electrical conductivity vs. Temperature. (**c**) Thermal conductivity vs. Temperature. (**d**) ZT vs. Temperature.

**Figure 5 materials-16-06961-f005:**
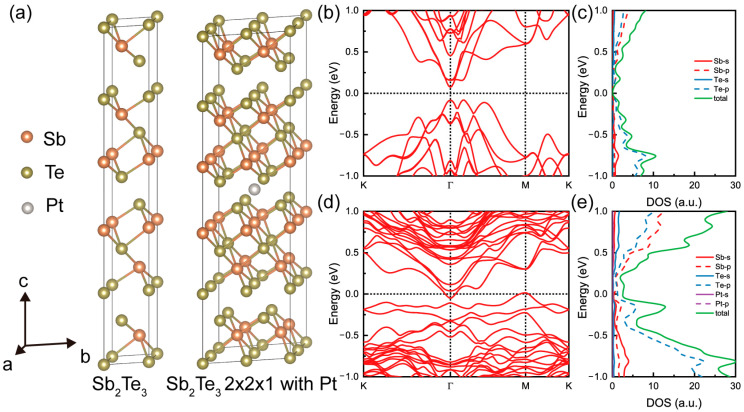
(**a**) Structural view of Sb_2_Te_3_ and Sb_2_Te_3_ 2 × 2 × 1 supercell with Pt. Orange, green, and gray balls represent Sb, Te, and Pt atoms. The calculated band structure with SOC for (**b**) Sb_2_Te_3_ and (**d**) Sb_2_Te_3_ 2 × 2 × 1 supercell with Pt, respectively. The calculated DOS with SOC for (**c**) Sb_2_Te_3_ and (**e**) Sb_2_Te_3_ 2 × 2 × 1 supercell with Pt, respectively.

**Figure 6 materials-16-06961-f006:**
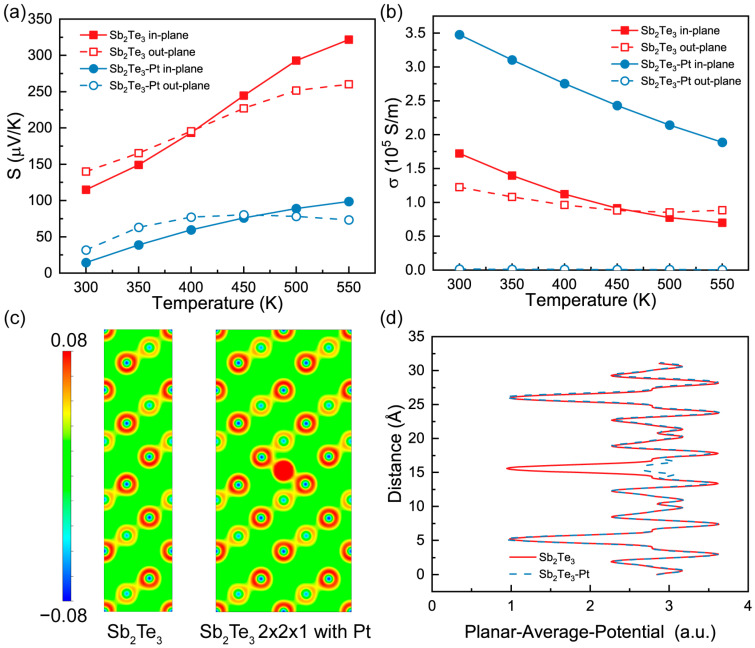
The calculated (**a**) Seebeck coefficients and (**b**) electronic conductivity as a function of temperature. The side view of (**c**) charge density and (**d**) planar average potential along c lattice.

**Table 1 materials-16-06961-t001:** The distribution ratio of Sb, Te, Pt elements.

Element	Mass (%)	Atom (%)
Sb	36.34	37.74
Te	61.25	60.70
Pt	2.41	1.56

**Table 2 materials-16-06961-t002:** Carrier concentration, carrier mobility, and electrical conductivity at 300 K.

Sample	P/N Type	Carrier Concentration (cm^−3^)	Carrier Mobility (cm^2^/V/S)	Electrical Conductivity (T = 300 K S/m)
ST-1	P	8.76 × 10^18^	4.63 × 10^2^	6.46 × 10^4^
ST-2	P	2.37 × 10^19^	3.63 × 10^2^	13.8 × 10^4^
ST-3	P	1.87 × 10^19^	2.79 × 10^2^	8.9 × 10^4^
ST-4	P	9.8 × 10^18^	2.5 × 10^2^	3.93 × 10^4^

## Data Availability

The data provided in this study could be released upon reasonable request.

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
