# Peer review of "Improved Thermoelectric Performance of Sb2Te3 Nanosheets by Coating Pt Particles in Wide Medium-Temperature Zone"

_materials, 2023, doi:10.3390/ma16216961_

Round 1

Reviewer 1 Report

Comments and Suggestions for Authors

In this manuscript, the authors explore the thermoelectric properties of Sb2Te3 coated with Pt nanoparticles. The dimensionless figure of merit reaches approximately 1.7 at around 600K in the 0.2wt% Pt sample, attributed to high electrical conductivity. Through theoretical analysis, the authors demonstrate an increase in the density of states near the Fermi level in a Pt-induced supercell structure. Consequently, they conclude that Pt doping is effective for enhancing thermoelectric performance.

I have the following comments on this manuscript:

(1) The manuscript is unclear about the role of 0.2 wt% Pt atoms in altering the electronic band structure. Figure 5 uses calculations based on about 1.6% Pt atoms in the supercell structure. If the percentage of Pt atoms is reduced to 0.2 wt%, would the band structure remain significantly altered? The authors need to justify the appropriateness of using a 2x2x1 supercell for their calculations. Also, the manuscript mentions an effective mass of 0.2875me in the 0.6 wt% case; explaining how this value differs from other cases would be helpful.

(2) The manuscript could benefit from an expanded discussion on electrical conductivity tensors and Seebeck coefficient tensors. Clarification is needed on the direction in which these properties are plotted in Figure 6, especially given that the BolzTraP software outputs these properties as tensors.

Based on the above, while the manuscript is informative for researchers in this field, I think that it is unsuitable for publication in Materials.

Author Response

The response is in the uploaded attachment.

Reviewer 2 Report

Comments and Suggestions for Authors

This question aims to gain a deeper understanding of the practical applications and implications of the research. 

1.Can you elaborate on the solvo-thermal and photo-reduction methods used to prepare Sb2Te3 nano sheets coated with Pt particles?

2. What specific thermoelectric properties were improved when Pt particles were added to Sb2Te3?

3. What are the implications of the sharp raise in the Density of States (DOS) for the material's electronic conductivity?

4. How was the ZT value determined, and why is it important in assessing thermoelectric materials?

5. Were there any challenges or limitations encountered during the research that should be considered when interpreting the results?

6. Can you provide more details on the room temperature and medium-temperature thermoelectric applications, and how this research contributes to these fields?

Comments on the Quality of English Language

Moderate editing of English language required

Author Response

(The authors gave the same response as above.)

Reviewer 3 Report

Comments and Suggestions for Authors

This manuscript describes the improvement in the thermoelectric performance of Sb2Te3 nanosheets by coating Pt particles.   After carefully reviewing the manuscript, I have observed a few areas with unclear information that require reconsideration. Following are my comments on the manuscript that needs to be seriously considered before possible publication.

1.     What wide medium-temperature zone is of target? and why? Providing clarity over the temperature range will help to understand its need, relevance, and importance.

2.     There is inconsistency while explaining and confirming about the doping and/or coating of Pt particles. While characterization results prove it as ‘coating’, the DFT study considered it as ‘doping’ which is contradictory. This must be further confirmed and clarified. It is imperative to note that coating is unlikely to introduce any changes to the band structure, while doping can significantly impact it.

3.     Page 2, lines 89-90 and 91: Explain how sheets dissolved first and then dispersed.

4.     Page 2, line 91: Pt nanoparticles are metallic, and how those are further reduced? explain by providing the chemical reaction for the same.

5.     How does SEM, and TEM analysis provide a comprehensive understanding of the thermoelectric performance? Page 4, line 159 and 184 needs to be rephrased.

6.     Introduction section: Elaboration on total thermal conductivity and their significance/impact will help to justify their importance in thermoelectric study.

7.     The term ternary chalcopyrite is not explained on page 1, line 43.

Recommendation

Minor revision.

Comments on the Quality of English Language

Minor editing and spell checks are required.

Author Response

(The authors gave the same response as above.)

Round 2

Reviewer 1 Report

Comments and Suggestions for Authors

The revised manuscript is clearer and the authors have correctly responded to all comments. Therefore, the revised manuscript is suitable for publication in Materials.

Reviewer 2 Report

Comments and Suggestions for Authors

The author has done all the corrections and added the suggested description in the revised manuscript. The manuscript can be accepted in the present form.